# Perceived impacts of COVID-19 and bushfires on the implementation of an obesity prevention trial in Northeast Victoria, Australia

**Jillian Whelan**[1,2], **Monique Hillenaar**[1]*, **Penny Fraser**[1], **Steven Allender**[1], **Michelle Jackson**[1], **Claudia Strugnell**[1,3], **Colin Bell**[1,2]

**1** Global Centre for Preventive Health and Nutrition, Institute for Health Transformation, Deakin University, Geelong, Victoria, Australia, **2** School of Medicine, Deakin University, Geelong, Victoria, Australia, **3** Institute for Physical Activity and Nutrition, Deakin University, Geelong, Victoria, Australia

* monique.hillenaar@deakin.edu.au

**Data Availability Statement:** The data collected in this study were collected from participants in a small rural community. If made available, this data

## Abstract

### Background

Calls for the adoption of a systems approach to chronic disease prevention date back at least ten years because of the potential to empower communities to identify and address the complex causes of overnutrition, undernutrition and climate change. Australia, like many countries, has high levels of obesity and extreme climate events. The Reflexive Evidence and Systems interventions to Prevent Obesity and Non-communicable Disease (RESPOND) trial aims to prevent unhealthy weight gain in children in 10 intervention and two pilot communities in north-east Victoria, Australia using community-based participatory approaches informed by systems science. Intervention activities co-designed in 2019 were disrupted by COVID-19 and bushfires. This paper explores the impacts of these 'shocks' on the local prevention workforce to implement actions within communities.

### Methods

A case study design involving one-hour online focus groups and an on-line survey (November 2021-February 2022). Purposive sampling was used to achieve diverse representation from RESPOND stakeholders including local council, health services, primary care partnerships and department of health. The focus group interview schedule and survey questions were based on Durlak and DuPre's implementation factors.

### Results

Twenty-nine participants from seven different communities participated in at least one of nine focus groups to discuss the impacts of COVID-19 and bushfires on localised implementation. Twenty-eight participants (97% of focus group sample) also completed the on-line survey. Implementation of RESPOND stalled or stopped in most communities due to bushfires and/or COVID-19. These shocks resulted in organisational priorities changing, loss of momentum for implementation, redeployment of human resources, culminating in fatigue

could identify participants of our study which would infringe our ethics approval for the study. These restrictions are imposed by the Deakin University Research Ethics Committee. Data will be made available upon request from Victoria Stead, Chair of the Deakin University Research Ethics Committee, via email (victoria.stead@deakin.edu. au) and from the Deakin University Research Ethics Committee via email (health-ethics@deakin. edu.au) for researchers who meet the criteria for access to confidential data.

**Funding:** RESPOND is funded by Australia's National Health and Medical Research Council's (NHMRC's) Partnership Projects Research Grants scheme, (GNT1151572) with further funding and in-kind contributions from 12 partner organisations who were signatories to the grant. Partners to the RESPOND grant were Deakin University (lead agency), the Victorian Government Departments of Education and Training and of Health and Human Services, Beechworth Health Service, Yarrawonga Health, Gateway Health, Numurkah District Health, Lower Hume Primary Care Partnership, Central Hume Primary Care Partnership, Upper Hume Primary Care Partnership, Goulburn Valley Primary Care Partnership, and VicHealth. Additional organisations who have joined the partnership since establishment include Greater Shepparton City Council, Murrindindi Shire Council, and Nexus Primary Health. The funders had no role in study design, data collection and analysis, decision to publish, or preparation of the manuscript.

**Competing interests:** The authors have declared that no competing interests exist.

and exhaustion. Participants reported adaptation of RESPOND, but implementation was slowed due to limited resources.

## Conclusion

Further research is needed to advance risk management strategies and protect resources within health promotion. System shocks such as bushfires and COVID-19 are inevitable, and despite multiple adaptation opportunities, this intervention approach was not 'shock proof'.

## Introduction

Childhood obesity impacts at least one in four children in Australia [1], with health implications both in childhood [2] and in later life [3]. The latest Cochrane review on childhood obesity prevention reported limited effectiveness of existing interventions and called for a systems approach to prevention, highlighting that interventions designed to target the complex drivers (e.g. multi-faceted and multi-level) of childhood obesity are likely to have the greatest impact [4]. The recent Lancet Commission on Obesity (2019) expanded the application of their 2011 call for a systems approach [5] to prevent obesity, to also prevent undernutrition and climate change [6].

In Australia, the total annual area of land affected by fire has increased significantly over the past 32 years [7], and in 2019 and 2020 Australia experienced fires in each of the six states and one of the two territories of Australia [8]. Northeast Victoria was one of the areas largely impacted by the 2019/20 bushfires, with the emergence of COVID-19 shortly following [9]. Varying degrees of public health orders were put into place, such as limitations on public gatherings (outdoor and indoor), closure of specified retail business types, working from home requirements, and limits to the physical distance people could travel from their homes [10]. Secondly, there were specific health consequences directly related to COVID-19. To date (August 2022), Australia has reported over 10 million cases of COVID-19 and more than 14,200 reported deaths [11]. Beyond the immediate disease, there were physical and mental health impacts on health workers [12] and documented impacts on mental health across community members more generally [13].

At the time that the 2019/20 bushfires and COVID-19 hit northeast Victoria, communities in this area were also participating in the Reflexive Evidence and Systems interventions to Prevent Obesity and Non-communicable Disease (RESPOND) research project. In 2018, the RESPOND trial commenced in the Goulburn Valley and Ovens Murray regions [14]. RESPOND is a 5-year, stepped-wedge cluster randomised controlled trial funded by the National Health and Medical Research Council of Australia and partners. It was scheduled to operate across five local government areas (step 1) from 2018, another five local government areas (step 2) from 2020 (2 years after step 1), and was preceded by work within two pilot communities from 2016. RESPOND uses community-based participatory research informed by systems science to build capacity within community stakeholders [15] who then co-design actions to prevent unhealthy weight gain and improve health-related quality of life in children aged 5–12 years. Co-design began with a series of three group model building (GMB) workshops [16] to build a systems map capturing local drivers of childhood obesity and preventive actions [16].

Project stakeholders commenced implementation of local prioritised actions from 2019. Examples of such actions included increasing healthy food access through shares and swaps,

food literacy and education programs, increasing opportunities for physical activity through enhancements to public spaces and come and try events, and increased communications and promotion of opportunities. The implementation for most communities was interrupted by bushfires in November 2019. The fires were described as 'unprecedented' [8], with 21% of Australia's temperate broadleaf and mixed forests biome burned (New South Wales and Victoria), compared with 2% typical of previous major fire years [17]. All communities in RESPOND were impacted to varying degrees by these bushfires, most directly through active fires, or smoke pollution. All RESPOND communities saw the reallocation of stakeholders, such as health promotion and council officers, to emergency response roles, where they were required to support the bushfire recovery and/or Victoria's COVID-19 response. In addition, these stakeholders along with their community members, were impacted by work disruptions, illness, and deaths associated with COVID-19.

Implementation of initiatives in real-world interventions is always challenging. The field of Implementation Science provides guidance to enhance this uptake of evidence into practice [18], and has identified specific factors for consideration during this implementation phase [19]. Implementation is likely to be even more challenging when communities are impacted by shocks such as bushfires and COVID-19. This manuscript explored the impact of bushfires and COVID-19 on the stakeholders' ability to implement RESPOND. This manuscript addresses the research question: 'what factors related to bushfires and COVID-19 impacted stakeholders' ability to implement RESPOND as planned?'

## Methods

### Design and theoretical frame

A case study approach was used to gather information on the impacts of bushfires and COVID-19 on the implementation of a large, funded trial. Ethics approval was obtained from Deakin University HEAG_H 172_2018. We drew on Community-Based Systems Dynamics (CBSD) and Implementation Science frameworks. CBSD informed the co-design of the RESPOND interventions and gave us a process for identifying and managing complex and dynamic community determinants of children's health. Implementation science, and the Durlak and Dupre [19] framework in particular, guided the development of the survey and focus group questioning. Following this, the framework also informed the theming and identification of implementation factors. CBSD was again used for summarising themes.

### Context–The RESPOND intervention

RESPOND is a community-led change initiative that focusses on creating healthier environments for children aged 0 to 12 years across 10 local government areas (approximately 30,000 children) in north-east Victoria, Australia. More detail is provided in the published protocol [14].

### Research team and reflexivity

Focus groups were conducted by Author 1 (PhD) as facilitator, with Author 2 (BHlthSc) as notetaker. Author 1 has specific training in qualitative methodologies. Author 2 has been involved in qualitative research previously. Author 1 was employed as a Postdoctoral Researcher and Author 2 as a Project Manager and Implementation Coach for RESPOND at the time of the research. Data collection and analysis was grounded in pragmatic epistemology. By adopting pragmatism, the researchers looked at understanding and addressing problems in a real- world context [20], embracing the messiness of everyday life [21]. Study insights were informed by team members' understanding of the context, the broader RESPOND study,

methodological experience, and field-based knowledge. Researcher perspectives were guided by the recent Lancet Commission on Obesity, community-based systems dynamics, and traditional health promotion frameworks such as the Ottawa Charter. At the time of conducting this research, all authors, except PF, had varying degrees of a pre-existing relationship with most participants through involvement with the RESPOND project. No other researchers or non-participants were present at the focus group sessions.

## Participants and recruitment

Participants were recruited via email through existing networks from two pilot communities (City of Greater Shepparton and Moira Shire) and five Step 1 communities (Strathbogie Shire, Murrindindi Shire, Mansfield Shire, Indigo Shire and City of Wodonga). Purposive sampling was used to achieve participation from a range of stakeholders involved with implementing RESPOND in each community. Invited participants (hereafter participants) included staff (community development and health promotion) from local government, the health sector, primary care partnerships, state government departments of health and sporting organisations. Written consent was provided by each participant. Where this consent was not received prior to the focus group session, verbal consent was sought and audio recorded at the beginning of this session, and written consent was received prior to analysis.

## Data collection

Data were collected from one-hour online focus groups and an on-line survey with RESPOND stakeholders (November 2021-February 2022). The focus group schedule (S1 File) and survey (S2 File) were based on implementation factors proposed by Durlak and DuPre [19] and used to facilitate the discussion of the impacts of COVID-19 and bushfires (henceforth 'shocks') on the capacity of RESPOND stakeholders to implement intervention activities in communities. Durlak and DuPre [19] reviewed and synthesised 542 implementation studies and identified five broad factors impacting implementation: community level factors (e.g., politics, funding); provider characteristics (e.g., perceived benefits of innovation); characteristics of the innovation (e.g., compatibility and adaptability); factors related to the prevention delivery system (e.g., work climate, shared decision making, leadership), and factors related to the prevention support system (e.g., training).

A focus group was scheduled for each of the Step 1 communities (n = 5) and one focus group for the pilot communities, to discuss the impact of the bushfires and COVID-19 on participants and the implementation of RESPOND. Additional focus groups (n = 3) were scheduled to accommodate requests to participate from people who had been invited but were unable to attend the initial sessions. Focus group transcripts were reviewed by authors 1 and 2 after each session. At the conclusion of the nine focus groups, authors 1 and 2 agreed that no new themes were emerging, and that data saturation was reached. Focus groups were audio and visual recorded using Zoom [22], and transcribed verbatim. A notetaker (author 2) took notes during each focus group as a backup and for clarification (if required) of the online recording. Once de-identified, transcripts were returned to all participants for review and editing. No significant feedback from participants was received; only simple grammatical and de-identifying advice. The finalised transcripts were imported into NVivo [23]. All identifiable information was securely stored via password protection on Author 2's Deakin University computer system.

At the conclusion of the focus group, participants were invited to complete a 15-question on-line survey. Responses to the questions were recorded on a 5-point Likert scale—Strongly Agree, Agree, Neutral, Disagree, Strongly Disagree. Results from the survey were exported to a

Microsoft Excel spreadsheet and summarised. This survey's purpose was to complement the qualitative data from the focus group questions.

## Data analysis

Transcripts were themed to the implementation factors identified by Durlak and DuPre [19] based on a codebook developed by authors 1 & 3 (S1 Table). An additional theme of 'system shocks' was added to capture the bushfire and COVID-19 events. Where aspects from the transcripts did not align with the framework, emergent themes were created. During the process of data analysis, the researchers most closely involved in data collection and analyses (Authors 1, 2 & 3) met regularly to discuss coding to the framework and the emerging themes, clarify analysis and share reflections. At this stage, input was sought from other team members with experience in the RESPOND project (Authors 4, 5, 6 & 7). De-identified quotes drawn from the thematic coding were used to illustrate themes. Summarised survey results were captured in a table, with each question grouped in to one of three themes. These themes were derived from the themes within the codebook (S1 Table). After analysis of the focus groups and surveys, the inter-relationship between the factors impacting implementation were connected visually using STICKE software [24]. This was done by Author 2 using CBSD principles and experience, and with support from systems science expert Andrew Brown. Connections were derived from the focus group discussions and were considered by all authors, to add value to the narrative and connections of the identified factors.

## Findings

A total of 29 people participated in one of nine focus groups. One participant attended two focus groups as they were representing a different community in each and provided input for each community. Twenty-eight of these participants also completed the survey. All identified stakeholder groups were represented by at least one participant, with highest participation from the health sector (n = 19), followed by local government (n = 5), primary care partnerships (n = 4), state government (n = 1), and regional sports assemblies (n = 1). See Table 1. All participants had been involved with RESPOND for at least two weeks, with most having been involved for the duration of the project to date (˜ 2 years). Preliminary findings were shared with participants through a regular network meeting and no significant feedback was provided.

## System shocks—bushfires and COVID-19

Participants described responding to bushfires as routine because of previous bushfire responses.

'. . .there's a historical aspect to this. The northeast of. . .Victoria has . . . every three or four years have dealt with major bushfires to varying levels of. . . extremity I suppose' (FG8, P1)

**Table 1. Participant characteristics.**

| Representation of participants | Number of participants |
|---|---|
| Health sector (Hospitals, Community Health) | 19 |
| Local government | 5 |
| Primary Care Partnerships | 4 |
| State Government (Department of Health) | 1 |
| Regional Sports Assembly (Sport North East) | 1 |
| **Total** | **30** |

However, the shock of COVID-19 was considered overwhelming because it was in addition to the bushfire shock and it impacted many more people. The sentiment, as captured in the quotes below, was that communities had faced one thing after another, and they were tired and perhaps less likely to engage with RESPOND as a result.

'...if it was just bushfires alone I can't imagine that would have had much of an issue, but it was a knock on effect, that plus, then COVID being the real impact.' (FG4, P2)

'...people were exhausted. They were fatigued, not just, you know, first of all, they were fatigued from bushfires, but then they got to COVID and they were still fatigued from bushfires, and then COVID on top of it.' (FG8, P2)

### Community level factors

**Policy.** Discussion of policy comprised two elements: firstly, there were elements of policy in terms of public health orders that limited, for example, stakeholder capacity to engage with community members in person; second, changes to state level health promotion guidelines were provided to local stakeholders for review during this period of COVID-19. The proposed changes had potential to impact the way stakeholders co-designed and prioritised their work. Participants reported concern of both the timing of this proposed policy change and the content which they interpreted as allowing less local context.

Regarding the proposed health promotion priorities to be funded, multiple participants expressed concern:

'The big elephant in the room, which we all know about, is the [new policy] guidelines. Um, they really, yeah, hinder us in the way that they're very prescriptive and there's not a lot of space for other things to happen.' (FG1, P3)

One participant noted that the policy guidelines could 'future proof' their work if they were aligned with community-led implementation:

'... if it's in a strategic plan, and there's alignment, and it has to happen, the turnover of the people is probably not as critical as it is with the linkages. Because if it's in a plan, it's got to be delivered, they're accountable for that. So, ... then it gets handed over to the next person and becomes that critical piece of work.' (FG8, P2)

Multiple participants reported transitioning, often with little warning, in and out of policy changes due to health orders responding to changing pandemic conditions (e.g. example community lockdowns). A consequence of this rapid change in policy priority meant practitioners were unable to plan with certainty any community engagement activities or implement the prioritised actions of RESPOND:

'I think so much of the impact as well is, what I said earlier, the stop and start nature of this year. I think we never were able to get momentum, or feeling of what we could do, you know, you'd have a few months of being able to go in person to see people and then we're back in lockdown, we're out of lockdown. It was too uncertain to plan and do things as well because it was so in and out this year.' (FG4, P3)

*'. . .we've since lost it again because of COVID. So, this continuous issue around building that momentum through community engagement to get community- led response, there's a massive barrier when disaster happens.' (FG8, P2)*

One focus group reported that after multiple efforts to restart an initiative, a decision was made to pause:

*'. . . we made the call as a team that we were going to stop at a point and just breathe and wait till we could get somewhat back together and go again, so we did that in May and of course we had another interruption, set of interruptions and everything since' (FG5, P1)*

**Funding.** While the funding for RESPOND did not change in response to the bushfires and COVID-19, participants reported that the shocks stretched in-kind contributions:

*'Upon reflection of how RESPOND was originally meant to fit into our current work role, the initial 10% that was sort of deemed as our EFT [equivalent full time] within our funding agreement, the 10% wasn't a realistic figure at all. We didn't have any additional funding or additional staff to be able to do it' (FG9, P2)*

This reflection on inadequate resourcing was experienced by others, who reported:

*'. . .we didn't have the resources. The amount of pre- preparation work prior to GMBs, the amount of work post GMBs, it was essentially, we had every single team member working on that, and doing the minimal amount of other programs as opposed to RESPOND. And it was very much, we piled in our EFT just for that. . . . . . So it was this ebb and flow of EFT to RESPOND. So, I don't think we had enough resources and we're identifying right now that we've got some staffing issues in [local area] and that we won't have enough for this year.' (FG8, P2)*

One community presented a more optimistic view, saying that the creation and funding of a RESPOND position enhanced community resilience (see related theme) following the COVID-19 and bushfire shocks and, protected RESPOND funding for its intended purpose in the face of competing priorities:

*'. . . [referring to person funded in position] having that role, be really pushing forward and keeping the momentum going, was definitely the reason that we were able to have the success that we've had' (FG7, P3)*

*'. . .we had [referring to colleague] employed with hours. . .so, for at least a day a week. . .we could quarantine those hours to RESPOND. . .so I think that helped in not diverting (colleague) to something else that could have been COVID-related.' (FG3, P1)*

## Provider characteristics

**Redeployment of resources (emergent theme–related to perceived need for innovation, self-efficacy and skill proficiency).** Many participants reported that their organisations asked

them to 'pivot' individual roles and responsibilities to *"bushfire recovery work"* (FG5, P1), *"testing"* and *"contact tracing"* (FG1, P3), and COVID-19-related *"communications"* (FG9, P2).

'. . . COVID also took a lot of our EFT for quite a while, at least from our organisation because so many different responses we were involved in, so for a good year, both myself and my co-worker for about half our jobs were probably doing COVID response, if not more' (FG4, P3)

This 'pivot' displaced health promotion work, including working on RESPOND.

'COVID's had a huge impact and especially on our facilitation team too because a lot of them, like, . . . myself was only one day a week, . . . I stopped doing prevention to just focus purely on COVID related work' (FG7, P1)

They also reported a lack of clarity from their organisations in terms of what tasks they were required to perform and when:

'. . .there was a long time when officially we weren't in the COVID response, but we suddenly had to man the front door, do screenings and stuff like that, and so it was a lot of unofficial "we just need bodies and we need hands to help as well", so there was a bit of giving in that way. It's only in the last six to nine months it's really become a much more official allocation of roles and workload. Prior to that, it was just grab as needed.' (FG4, P3)*

In other organisations, the impact of redeployment was not as pronounced, but time available for health promotion was still impacted because of physical distancing requirements and 'work-from-home' advice:

'. . .that's not really been the cause of our lack of work in the RESPOND space, it's more about that community engagement and that lack of ability to engage in a normal, in inverted commas, way, yeah so it's not been a lack of staff, it's more been about a lack of ability to get out and about and do the things in the way that we usually do, do things.' (FG4, P2)*

### Characteristics of the innovation

**Compatibility (contextual appropriateness, fit, congruence, match).**   A key finding was that participants reported community priorities being pushed away from the objectives of RESPOND and towards managing COVID-19:

'. . .there was a lot of momentum and there was a lot of people that were passionate about the project and willing to be a part of it, . . . that have just been pulled other places and that have other priorities like we all have throughout COVID. And, you know, this [RESPOND] has been put on the backburner a bit' (FG7, P2)*

Participants described how the shocks made some priorities more urgent than others. Children's healthy eating and physical activity was less of a priority due to COVID-19 despite the planning, agreements, partnerships and commitments that were in place. Instead, the emphasis was on mental health:

*'All of our Planned stuff has been put on the back burner to deal with COVID requirements. . .
and majority of our conversations with our stakeholders has been around mental health.'
(FG6, P1)*

**Adaptability (program modification, reinvention).** Participants reported that adapting
to COVID-19 made delivery of RESPOND harder because of requirements to work from
home and to physically distance. This meant that meetings could not be face- to- face. Moving
online was initially seen as a positive because it opened the possibility of engaging with people
who had previously been difficult to reach, such as those living out of town. This did not even-
tuate for RESPOND however:

*'. . .you have to have some really committed people for them to actually be able to travel to do
face-to-face. So I initially thought "Oh, this is going to be great, get lots of people from across
the [local government area]" [into a virtual platform], but we've actually really struggled to
engage more broadly across the [local government area]' (FG9, P2)*

Participants described engaging with community members online as a poorer quality of
engagement compared to face- to- face, and said it impacted their ability to deliver RESPOND.

*'. . .through COVID, one of the biggest things was that there is this new expectation around
agility, and that adaptability, . . . and using digital engagement to actually get things done,
but the thing is, there's a difference between digital engagement and community engagement,
they're very different things, and you cannot necessarily do community engagement digitally.
And we had very poor uptake when we tried to flip it.' (FG8, P2)*

This online shift may have particularly impacted RESPOND because initial community
engagement was face- to- face and cumulative, enabling multiple formal and informal discus-
sions that some participants felt did not transfer well to the online environment.

*'. . . because RESPOND was built on, especially here, a face-to-face model, you know, to
take that offline, that's just not realistic really,' (FG5, P1)*

*'. . . it's all well and good to try to take things online, but is it actually having the same
impact that we want it to have? Is it having the same meaning? Are we actually still reaching
the purpose of the project in the first place by going online?' (FG6, P6)*

One participant outlined reasons they considered online engagement inferior:

*'I felt like with the group of people and it's, it happens with all online things. . .people become
hesitant to interrupt, . . . you don't have those organic natural conversations, . . . I just feel like
online you do miss out on some of the body language or cues or people that may want to say
something but are too scared to interrupt online.' (FG9, P3)*

Conversely, one community reported that the move to online engagement wasn't detrimen-
tal because of the foundation that was built prior to the first experience of lockdowns:

*'so we did, obviously did all of our's [referring to GMBs] just prior to the bushfires. So, which
was probably really good for [our community]. . ..we had our group of interested community
people, and then the other thing that did happen is that we offered our first lot of face- to- face*

*interviews from our groups after bushfires, which was sort of end January pre- entering any form of COVID restrictions, so we did meet, we got one meeting with each of our groups. . . which I think, then, when we then had to go to online meetings, we had some really good ideas that had come out of those focus groups, as well as the meeting, to start to move forward to get some runs on the board.' (FG3, P1)*

## Factors relevant to the prevention delivery system: Organisational capacity

**Integration of new work (extent to which an organisation can incorporate an innovation in its existing practices).** Schools transitioning to remote learning and the restriction of early childhood services to essential workers (e.g., health care staff, police), pushed RESPOND activities further aside:

*'. . .unfortunately, by February when everybody was sort of back on board and raring to go, a lot of actions were either in community or working with settings such as schools, early years services, maternal child health nurse, health services that just were bombarded when it came to COVID.' (FG7, P1)*

*'it's been hard to get into, get involved with our stakeholders, especially schools because they've been so busy and impacted.' (FG6, P3)*

**Shared decision making (local input, community participation or involvement, local ownership collaboration).** Participants described both bushfires and COVID-19 severely hampering their ability to undertake community participation activities, to build local ownership of RESPOND, and strengthen collaborations as originally intended.

*'. . . there's being really minimal contact from, like between us and the community, just because of yep bushfires, COVID . . .' (FG1, P3)*

*'. . . as with probably everybody, COVID hit and yeah the wheels fell off effectively. . . . . .We still have been doing little bits and pieces in this space but yeah, certainly not anything to the scale of where we were hoping to get to after that last GMB with that great engagement' (FG4, P2)*

This was particularly true in schools that would normally be a core partner in health promotion activities.

*'. . . it's been hard to get into, get involved with our stakeholders, especially schools because they've been so busy and impacted.' (FG6, P3)*

**Leadership and champions.** Participants described previous experiences of bushfires, where local 'champions' had agreed to lead community prevention work but were pulled away to the urgent tasks of bushfire recovery. They expressed frustration that this had occurred again and that they were still vulnerable to shocks.

*'. . . we get stuck in these cycles where we'll have a bushfire, and that impacts on everyone's ability because you get staff burnout, you get people that you know, like I said, just have to*

concentrate on themselves and their own issues and they can't, you know, be champions' (FG7, P1)

*'Our key backbone group that we were going to utilise and lean on a lot was going to be our Rotary Club, and then our Rotary Club did go [and] give assistance to the bushfire area because (referring to local community) was obviously quite close to some of the other areas impacted. So we sort of lost a bit of engagement with them. . .' (FG4, P3)*

Participants noted that leadership changes are exaggerated in rural communities compared to metropolitan communities because leaders have multiple roles:

*'. . . because of the fact that we had community members there that wore so many hats. . .. Going into bushfires, . . . they were also involved in the CFA and so you know had no time to deal with all of the RESPOND stuff' (FG1, P3)*

**Fatigue (emergent theme).**　Many comments were made by participants regarding the uncertainty and exhaustion that both staff and community felt resulting from bushfires and COVID-19:

'Within the community and their capacity, I think everybody is just exhausted from the past two years and they're really struggling to take on anything extra' (FG4, P1)

*'They were so sick of Zoom. They were so sick of cancelling, you know, the emotional attachment and the financial attachment to set things up and cancel. In the end there's been a lot of "Can we just wait 'til this is over?"' (FG6, P2)*

Participants also described fatigue in community stakeholders, particularly schools.

*'. . .it's also the pushback from community, especially at the moment people are still getting back on their feet. Schools are still trying to get kids back in classrooms.' (FG1, P3)*

**Resilience and re-engagement** *(emergent theme).*　Aware that original actions needed to be reviewed, several RESPOND community facilitators sought to re-engage their communities to re-prioritise and potentially identify new priorities. The quotes below demonstrate an understanding of the need to reassess community needs and re-engage with them. Evidence was seen of resilience and commitment of facilitators:

*'I think it's almost more important now than ever to be getting that community's perspective of what they need, because it's so varied, even within the one LGA (local government area) it's quite varied what our little towns need in regard to support and yeah, their requirements, so I think it's more important now than it was before.' (FG6, P1)*

There was also evidence of community facilitators being aware of the need to honour people's initial engagement and provide them with re-engagement opportunities. The questions posed in the below quote indicate strategies for adaptation to the new landscape the community were facing. They also demonstrate an understanding for the importance of planning to ensure actions could be operationalised:

*'the people that we have engaged, I think it's really important to go back and give them the opportunity to revisit what those actions were. What is now going to work in our community? Who's involved? Are there new leaders? Are there new people who are influencing people within the community?' (FG9, P2)*

## Factors related to the prevention support system

In terms of the training needs of our participants, most acknowledged they had the skills required to undertake their roles. Participants reported resilience and an ongoing commitment to the health of local children and community wellbeing. What was less clear, was how to engage and successfully deliver a version of RESPOND in the wake of these system shocks.

*'I think COVID has, and that break and bushfires, has put everyone's energy into a different space. And I think it was really important for us to do part two [revisit priorities for RESPOND] for our community to say this space is actually really, really important despite COVID, or alongside COVID, healthy eating/ physical activity is really important for our health, to keep ourselves healthy during that time of the pandemic as well.' (FG9, P2)*

*'all of the way we look at the world and work and how we operate has had to shift, you know, . . . what we thought might happen or what we thought might be possible, and what we could do, you know, all had to get thrown out the window and to re- look at what does this mean. . .' (FG2, P2)*

In reflecting on the application of their skills in the use of community-based system dynamics, participants reflected that they were able to implement such work, develop a shared understanding of the issues, and commit to moving forward. Participants also reflected on the skills and shared values inherent in their rural communities:

*'. . .it's easy-ish to identify lots of shared values, you know, working together because we've got lots of shared values and I think, I don't know, that felt to me like it was highlighted in COVID a bit. Like, there's a, community looking after community, community caring about each other's wellbeing. For example, our local supermarket. . . bought a van, so it started home delivery during COVID, . . . you know, there were people on Facebook, on the noticeboard saying, you know, "I'm going shopping if anyone wants anything"' (FG6, P4)*

These qualitative findings are expanded through the inclusion of the survey data, that provided more detail to some of the themes.

## Survey

The majority of the 28 participants in the survey agreed/strongly agreed that their workplace considered RESPOND relevant to local needs (75%), was committed to the shared vision of RESPOND (89%), and that RESPOND was adaptable to the needs of their workplace (72%) (Table 2). They also agreed/strongly agreed that strong community partnerships existed (78%). However, only 29% agreed/strongly agreed that their organisation had enough funding or could reorientate funding towards RESPOND, and only 36% agreed/strongly agreed that they had clear roles and responsibilities within RESPOND. Regarding political support, support from local political figures was high (67%), but support from state level politics was less

**Table 2. Survey results.**

| Statement | RESPONSE n(%) | | | | |
|---|---|---|---|---|---|
| | Strongly Agree | Agree | Neutral | Disagree | Strongly Disagree |
| **Workplace/ Organisational- level factors** | | | | | |
| My workplace considers RESPOND relevant to local needs. | 12 (43) | 9 (32) | 5 (18) | 2 (7) | 0 (0) |
| Our organisation has enough funding/can re-orient funding to meet the needs of RESPOND. | 1 (4) | 7 (25) | 8 (29) | 11 (39) | 1 (4) |
| Our workplace/team members have the skills needed to run group model building. | 1 (4) | 16 (57) | 4 (14) | 6 (21) | 1 (4) |
| Our workplace/team members have the skills and connections needed to implement RESPOND actions. | 3 (11) | 20 (71) | 2 (7) | 3 (11) | 0 (0) |
| There is a positive work climate in our organisation. Morale, trust, collegiality and dispute resolution is strong. | 9 (32) | 14 (50) | 3 (11) | 2 (7) | 0 (0) |
| RESPOND is adaptable to the needs of our workplace. | 5 (18) | 15 (54) | 8 (29) | 0 (0) | 0 (0) |
| We/our workplace(s) have a strong commitment to a shared vision that aligns with the vision of RESPOND i.e. to co-create community-led change to improve the health of children. | 13 (46) | 12 (43) | 2 (7) | 1 (4) | 0 (0) |
| **Community–level factors** | | | | | |
| My community considers RESPOND relevant to local needs. | 4 (14) | 12 (43) | 7 (25) | 4 (14) | 1 (4) |
| The prevention theory and research component of RESPOND fits our community priorities and way of working. | 4 (14) | 19 (68) | 5 (18) | 0 (0) | 0 (0) |
| Our community partnerships are strong. | 6 (21) | 16 (57) | 5 (18) | 1 (4) | 0 (0) |
| RESPOND is adaptable to the needs of our community. | 10 (36) | 15 (54) | 3 (11) | 0 (0) | 0 (0) |
| **Policy, politics, planning, resourcing (Other)** | | | | | |
| Local politics (mayor/council/local govt reps) are supportive of RESPOND. | 2 (7) | 17 (61) | 6 (21) | 3 (11) | 0 (0) |
| State policy supports the approach of RESPOND. | 1 (4) | 12 (43) | 6 (21) | 8 (29) | 1 (4) |
| We have strong strategic planning, clear roles and responsibilities within RESPOND. | 0 (0) | 10 (36) | 7 (25) | 10 (36) | 1 (4) |
| We have adequate access to technical assistance, training, skills development within RESPOND. | 3 (11) | 14 (50) | 8 (29) | 2 (7) | 1 (4) |

clear at 47% of participants who agreed/strongly agreed that state policy supports the approach of RESPOND. See Table 2.

Fig 1 demonstrates the use of CBSD to analyse the factors revealed in this study interacting with bushfires and COVID-19. The colours identify the five Durlak and DuPre factors which are listed in the legend. A solid arrow between two factors demonstrates a relationship travelling in the same direction ie: as 'shocks in the system' increase, 'fatigue' also increases. Likewise, 'as shocks in the system' decrease, 'fatigue' also decreases. A dotted arrow indicates that there is a relationship travelling in opposite direction ie: as 'fatigue' increases, 'engagement' decreases, and as 'fatigue' decreases, 'engagement' increases. There are some examples in this figure which demonstrate a feedback loop, which recognises that some factors can get caught in a cycle ie: As 'fatigue' increases, 'engagement' decreases, which also decreases 'momentum'. This study found that with a decrease in 'momentum', 'fatigue' was in fact increased, which was seen with the stop-start nature of COVID restrictions.

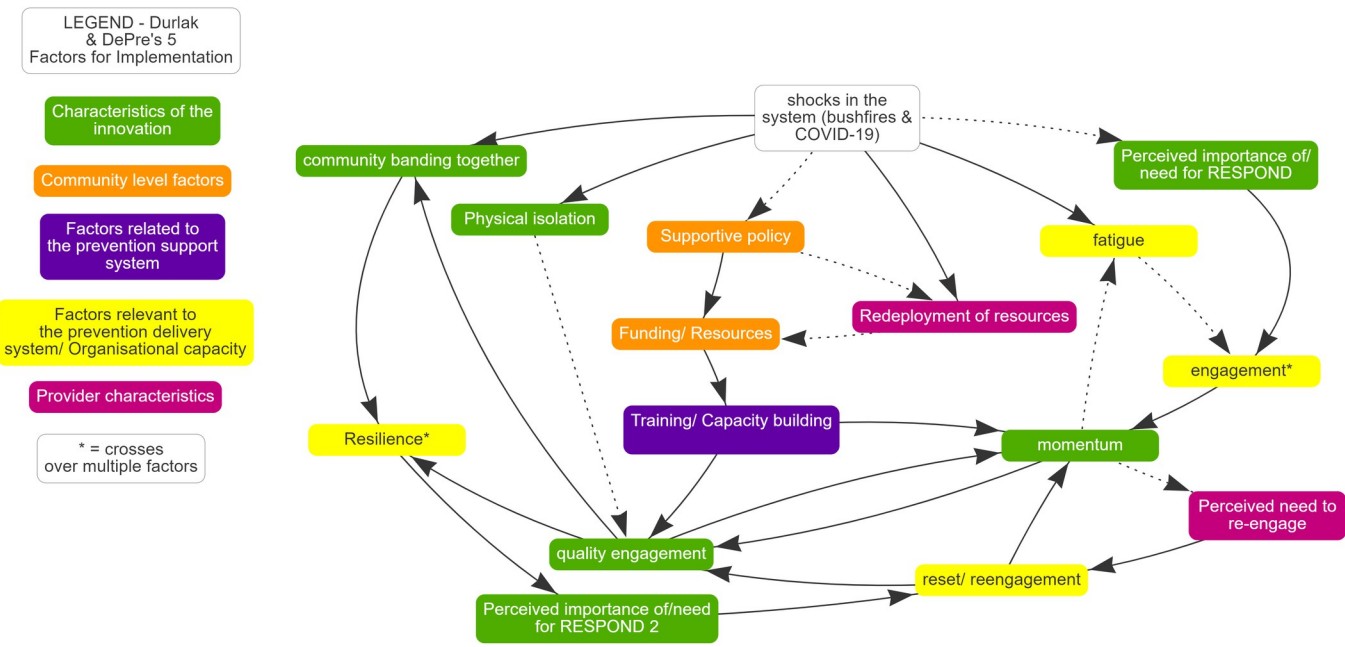

**Fig 1. Visual depiction of focus group themes and how they interact with the shocks in the system.**

## Discussion

### Main findings

The implementation of a large, community-based trial to prevent childhood obesity in regional Victoria, Australia was heavily impacted by COVID-19 and further compounded by bushfires. Key factors associated, included the impact of unexpected funding and policy changes within communities and at the provider level, staff churn, redeployment to engage with the COVID response, limitations of working in online environments, changing priorities for leaders, and fatigue across all levels due to the intensity of the COVID response. Even with these shocks, participants also reported resilience, adaptability and an ongoing commitment to the goal of RESPOND in improving the health and wellbeing of children.

### Comparison to literature

High redeployment levels of health promotion and community development staff to address COVID-19 has been observed across health sectors in Australia and internationally. Studies have reported similar impacts in low and middle-income countries, though perhaps even deeper where there was a rapid reduction in basic maternal, neonatal and child health services, indicating limited resilience within health systems [25]. Other studies report the finding here that community health workers were mobilised to assist with the COVID-19 response [26] and these roles changed rapidly, often without clear guidance or training. The impact of this redeployment and re-prioritisation has been variously described as affected [26], compromised [25], eroded [27] and stressed [28].

The re-deployment of existing staff to assist with the pandemic meant that multiple core roles were left unattended. While specific efforts were made to ensure RESPOND took an adaptive and reflexive approach [14], the size of the shock meant human resources were removed from their core roles and RESPOND work was paused until COVID passed, meaning momentum was stymied.

While staff priorities shifted, so did that of the community. It was reported that many community members were not able to prioritise the work of RESPOND as they were being pulled in other directions and began to have more focus on addressing the mental health impacts of these shocks on communities. COVID-19 has been reported to increase psychological distress, and adversely impact mental and physical health among others [29].

This study provided a reminder that active steps need to be taken by health promoters to generate and sustain political will for prevention [30], or community-based innovations like RESPOND may be inadequately resourced to maintain core business in the face of system shocks, as was the case in Queensland from 1984 to 2014 [31].

In this study, online engagement was reported as being less effective and less appropriate than in-person community engagement. These findings contrast with other studies, which found participants benefitting from online engagement and a broader representation of communities reached [32, 33] Given the growth in digital health technologies and potential of online technologies to reach marginalised community members, this is an area that warrants further research.

Inadequate resourcing of health promotion work has been a long- standing issue [34], and these findings suggest that this was amplified by the disruptions of bushfires and COVID-19 in Northeast Victoria. Consistent with findings of the thematic analysis by Kavanagh et al. [35] who interviewed policy and program administrators in public health, and local community workers and volunteers in Tasmania, Australia, this study identified the importance of valuing people as a resource, respecting relationships both with community and organisations, and having adequate funding. The importance of soft infrastructure, such as relationships and networks were identified, both by Kavanagh [35] and in this study. It was also identified, that the stronger the soft infrastructure in combination with adequate resources, the more likely those communities were able to cope with shocks, as seen by one RESPOND community.

## Strengths and weaknesses

This is one of only a handful of studies to explore the impact of external shocks to intervention delivery. Participants were experienced practitioners directly involved in the delivery and support of health promotion. Qualitative, quantitative and systems science (CBSD) methods were used to analyse the results. The lack of varied characteristics of participants was one limitation of the research. School representatives were missing from this study, and due to significant staff turnover in the community health and community development workforce over this period, there were many people involved with RESPOND who had changed employment. Two participants who were no longer engaged with RESPOND still shared their insights within the focus groups.

Interpretation of any focus group data is subject to personal bias and in the case of this research, pragmatism, where the researchers were looking for the practical consequences of the bushfires and the COVID-19 pandemic. It is possible that with this pragmatic lens, some nuance is lost.

## Conclusion

Populations would benefit from a better funded and more prominent health promotion workforce that could focus on their role in implementing funded trials and preventing disease longer term. Health policy support that prioritises prevention could assist this. Without fundamental changes to policy and funding, health promotion practice is always at risk of the next 'shock' where resources will again be 're-deployed' and their core skills undervalued.

Research should include adaptive methods rather than be prescribed, so that work can continue when shocks occur.

Future research should explore additional methods that acknowledge complexity when working with shocks and how to track these shocks in real time. Additional resources required in health promotion roles needs to be identified to ensure that even when pivoting is necessary, core business can continue. The next step for this research is to reflect on what stakeholders consider could 'shockproof' their work.

Bushfires and COVID-19 provide an ongoing opportunity to learn how to harness prevention work and funded trials though adaptation and efficient resource allocation. This study identified that health promotion expertise was seen as 'flexible', rather than essential to the core business of prevention. When 'shocks' occur, the implementation of preventative approaches like RESPOND are reduced in priority to enable a crisis response.

## Supporting information

**S1 File. Focus group script.**
(PDF)

**S2 File. Survey questions.**
(PDF)

**S1 Table. Theme codebook.**
(PDF)

**S2 Table. Completed COREQ checklist.**
(PDF)

## Acknowledgments

The authors express gratitude to Mr Andrew Brown for assistance in developing Fig 1 and sourcing references. They also thank all stakeholders who participated in this study.

## Author Contributions

**Conceptualization:** Jillian Whelan, Steven Allender, Claudia Strugnell, Colin Bell.

**Data curation:** Jillian Whelan, Monique Hillenaar, Penny Fraser, Michelle Jackson.

**Formal analysis:** Jillian Whelan, Monique Hillenaar, Penny Fraser.

**Funding acquisition:** Steven Allender, Claudia Strugnell, Colin Bell.

**Investigation:** Jillian Whelan, Monique Hillenaar.

**Methodology:** Jillian Whelan, Monique Hillenaar.

**Project administration:** Jillian Whelan, Monique Hillenaar, Claudia Strugnell.

**Resources:** Jillian Whelan.

**Software:** Steven Allender.

**Supervision:** Jillian Whelan, Claudia Strugnell, Colin Bell.

**Validation:** Jillian Whelan, Monique Hillenaar, Penny Fraser.

**Visualization:** Jillian Whelan, Monique Hillenaar.

**Writing – original draft:** Jillian Whelan, Monique Hillenaar, Penny Fraser, Claudia Strugnell.

**Writing – review & editing:** Jillian Whelan, Monique Hillenaar, Penny Fraser, Steven Allender, Michelle Jackson, Claudia Strugnell, Colin Bell.

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
