## [Decision Letter · Decision Letter 0]

14 Nov 2022

PONE-D-22-25022The impact of COVID-19 and bushfires on implementation of an obesity prevention trial in Northeast Victoria, AustraliaPLOS ONE

Dear Dr. Whelan,

Thank you for submitting your manuscript to PLOS ONE. After careful consideration, we feel that it has merit but does not fully meet PLOS ONE’s publication criteria as it currently stands. Therefore, we invite you to submit a revised version of the manuscript that addresses the points raised during the review process. Please submit your revised manuscript by Dec 28 2022 11:59PM. If you will need more time than this to complete your revisions, please reply to this message or contact the journal office at plosone@plos.org. Please include the following items when submitting your revised manuscript:A rebuttal letter that responds to each point raised by the academic editor and reviewer(s). You should upload this letter as a separate file labeled 'Response to Reviewers'.A marked-up copy of your manuscript that highlights changes made to the original version. You should upload this as a separate file labeled 'Revised Manuscript with Track Changes'.An unmarked version of your revised paper without tracked changes. You should upload this as a separate file labeled 'Manuscript'.

We look forward to receiving your revised manuscript.

Kind regards,

Maria Elisabeth Johanna Zalm, Ph.D

Editorial Office

PLOS ONE

Journal Requirements:

 "RESPOND is funded by Australia’s National Health and Medical Research Council’s (NHMRC’s) Partnership Projects Research Grants scheme, (GNT1151572) with further funding and in-kind contributions from 12 partner organisations who were signatories to the grant. Partners to the RESPOND grant were Deakin University (lead agency), the Victorian Government Departments of Education and Training and of Health and Human Services, Beechworth Health Service, Yarrawonga Health, Gateway Health, Numurkah District Health, Lower Hume Primary Care Partnership, Central Hume Primary Care Partnership, Upper Hume Primary Care Partnership, Goulburn Valley Primary Care Partnership, and VicHealth. Additional organisations who have joined the partnership since establishment include Greater Shepparton City Council, Murrindindi Shire Council, and Nexus Primary Health. "

   "NO COMPETING INTERESTS TO DECLARE"

Additional Editor Comments:

Your manuscript has been assessed by two peer-reviewers and their reports are appended below. 

The reviewers comment that your manuscript could be strengthened with additional detail and clarification, as well as a refocus of the introduction so that it includes more detail on the activities involved in the RESPOND intervention. In addition, one of the reviewers states that some of the conclusions drawn do not seem to stem from the data gathered, or at least reported in this study.

Could you please carefully revise the manuscript to address all comments raised?

Reviewers' comments:

Reviewer's Responses to Questions

**Comments to the Author**

1. Is the manuscript technically sound, and do the data support the conclusions?

Reviewer #1: Partly

Reviewer #2: Yes

2. Has the statistical analysis been performed appropriately and rigorously? 

Reviewer #1: N/A

Reviewer #2: Yes

3. Have the authors made all data underlying the findings in their manuscript fully available?

Reviewer #1: No

Reviewer #2: Yes

4. Is the manuscript presented in an intelligible fashion and written in standard English?

Reviewer #1: No

Reviewer #2: Yes

5. Review Comments to the Author

Reviewer #1: This paper discusses the implementation experience of study team members in the context of bushfires and COVID-19. Overall, the paper would benefit from more clear and concise writing, particularly the introduction and the discussion.

Title

1. I recommend changing the title to something that more specifically describes the qualitative nature of the study. Perhaps “Perceived impacts of…” or “Facilitators and barriers to implementation of…”

Introduction

1. Overall, I find the introduction to be too long, with excessive background on the problem of obesity and bushfires, and not enough background on what activities are actually involved in the RESPOND intervention. It would be helpful to have a figure or table that lays out more detailed examples of projects/prioritized actions taken by some of the community stakeholders in their communities. The description as it is, remains very abstract and it is difficult to get a full sense of their work within RESPOND.

Methods

1. Line 115: what is a Step 1 community?

2. Line 160: There should be a reference to Figure 1 here.

Results

1. Line 180-206: I find it difficult to follow this paragraph. Who is issuing the proposed policy changes that were provided to local stakeholders for review? What were some of these policy changes? What are the “new policy guidelines” that the participant is referring to as the big elephant in the room?

2. Line 304: What is LGA?

3. Line 434: This portion of the paper would benefit from a sentence or two to help with transition to the next section, rather than just ending on a quote.

Discussion

1. Line 476-480: This particular conclusion does not seem to stem from the data gathered. Did some participants discuss policy changes that seemed to occur rapidly with political will?

2. Lines 481-487: The two examples given to cite benefits of online engagement are very specific and seem quite unrelated to the work that the RESPOND team was trying to do online. Are there more relevant examples in the literature?

3. Lines 505-end: All of the text after the conclusion reads like an unfinished outline, rather than the end of a scientific manuscript. Perhaps there is a PLOS ONE style that I am not familiar with but I find it difficult to follow..

Table 1

1. I recommend removing the “N” from each table cell and just noting “n(%)” at the top of the table to indicate what format the data is presented in.

Figure 1

1. The figure is blurry and contains a lot of information that is not explained. It should have a legend that walks the reader through each piece of the figure and how to interpret what it is representing.

Reviewer #2: Great work - this is a very important and unique topic. I have not thought about how bushfires might impact upon treatment for obesity before. Given Australia's obesity epidemic and the sheer number of people that the bushfires impacted upon, this is likely a topic that affects many people. I thought that your discussion around online support for obesity not being well liked (as opposed to online treatment for Diabetes) to be particularly useful for clinicians who design interventions for people who are obese. The methods of the article appear sound. There are some minor language errors, please do a further proofread of the article.

6. PLOS authors have the option to publish the peer review history of their article (what does this mean?). If published, this will include your full peer review and any attached files.

Reviewer #1: No

Reviewer #2: No

---

## [Author Response · Author response to Decision Letter 0]

28 Feb 2023

Thank you for the opportunity to submit a revised manuscript to PLOS ONE. Please see 'Response to Reviewers' document uploaded with this submission for more information on how we have addressed reviewer comments.

---

## [Decision Letter · Decision Letter 1]

21 Mar 2023

PONE-D-22-25022R1Perceived impacts of COVID-19 and bushfires on the implementation of an obesity prevention trial in Northeast Victoria, AustraliaPLOS ONE

Dear Dr. Whelan,

Thank you for submitting your manuscript to PLOS ONE. After careful consideration, we feel that it has merit but does not fully meet PLOS ONE’s publication criteria as it currently stands. Therefore, we invite you to submit a revised version of the manuscript that addresses the points raised during the review process.

One of the original reviewers has reviewed the revised manuscript and is happy with the changes made. I have recently taken on the manuscript as Academic Editor and I have suggested a number of revisions (below) that are necessary before it can be published in PLOS One. Please take these on board as you revise your manuscript.

We look forward to receiving your revised manuscript.

Kind regards,

Elizabeth McGill

Academic Editor

PLOS ONE

Additional Editor Comments:

Introduction:

- Please revise the structure of paragraph 2 because it currently goes from wildfires, to COVID, to RESPOND and then back to COVID; it would make more sense to move the sentence starting “Communities in this area…” somewhere else so that it does not interrupt the flow of the section about COVID

- The paragraph starting “Project stakeholder commenced implementation …” lacks specificity: when did actions started?; when did the bushfires occur? What type of stakeholders were reassigned?

Methods:

- In the section ‘Design and theoretical frame’, more detail is needed here on the how theory influenced the research. How did implementation and systems sciences influence the research?

- What was the focus of the survey? Please provide a description in text and the full set of questions in the supplementary material (with the focus group topic guide); please also explain the rationale for conducting the survey.

- There is only one sentence on how the inter-relationships were visualized; this process needs more detail, e.g. which author(s) made decisions about the connections, was the work informed by any specific systems science method?

Findings:

- Please include a table of participant characteristics

- Throughout, I wanted more of the research team’s interpretation and contextualisation, as currently written, many of the quotes are left to speak for themselves without much clear analysis from the research team

- I think the study would be stronger if the focus group and survey data were integrated, rather than being reported entirely separately.

Discussion:

- Strengths and limitations: please revise to write in paragraph form, rather than a list of bullet points. The authors says systems science methods informed the analysis, but this is not described anywhere in the manuscript and it’s not clear the extent to which systems science informed the work (except the occasional mention of a concept from systems thinking like ‘feedback loop’ or ‘adaptation’)

Conclusion (in the text):

- Sentence “When ‘shocks’ occur, the implementation of preventative approaches like RESPOND are 544 abandoned for a crisis response.” – is “abandoned” in line with the findings? Perhaps consider the wording

Conclusion (in the abstract)

- “our systems approach was not ‘shock proof’” – I do not follow this from the results. Do you mean the intervention’s approach or the research approach? If systems shocks are inevitable, then wouldn’t you expect that no intervention could be shock proof? This needs revising

- It is helpful to have research recommendations, but the manuscript should end with a conclusion about the findings

Please include a COREQ checklist as supplementary material

Reviewers' comments:

Reviewer's Responses to Questions

**Comments to the Author**

1. If the authors have adequately addressed your comments raised in a previous round of review and you feel that this manuscript is now acceptable for publication, you may indicate that here to bypass the “Comments to the Author” section, enter your conflict of interest statement in the “Confidential to Editor” section, and submit your "Accept" recommendation.

Reviewer #1: All comments have been addressed

2. Is the manuscript technically sound, and do the data support the conclusions?

Reviewer #1: Yes

3. Has the statistical analysis been performed appropriately and rigorously? 

Reviewer #1: Yes

4. Have the authors made all data underlying the findings in their manuscript fully available?

Reviewer #1: No

5. Is the manuscript presented in an intelligible fashion and written in standard English?

Reviewer #1: Yes

6. Review Comments to the Author

Reviewer #1: The authors have done a nice job addressing my areas of concern. The table and figure are now easier to interpret, the introduction sets up the appropriate background, and the methods are well explained.

7. PLOS authors have the option to publish the peer review history of their article (what does this mean?). If published, this will include your full peer review and any attached files.

Reviewer #1: No

---

## [Author Response · Author response to Decision Letter 1]

30 May 2023

Thank you for the opportunity to further edit and improve this manuscript. Please see attachment provided (Response to Editor) with this submission, which provides detailed responses to the suggested adjustments.

---

## [Editor Report · Decision Letter 2]

6 Jun 2023

Perceived impacts of COVID-19 and bushfires on the implementation of an obesity prevention trial in Northeast Victoria, Australia

PONE-D-22-25022R2

Dear Dr. Hillenaar,

We’re pleased to inform you that your manuscript has been judged scientifically suitable for publication and will be formally accepted for publication once it meets all outstanding technical requirements.

Kind regards,

Elizabeth McGill

Academic Editor

PLOS ONE

Additional Editor Comments (optional):

Thank you for revising your manuscript in line with the reviewer and my comments. I look forward to seeing this published.
---

## [Editor Report · Acceptance letter]

9 Jun 2023

PONE-D-22-25022R2 

Perceived impacts of COVID-19 and bushfires on the implementation of an obesity prevention trial in Northeast Victoria, Australia 

Dear Dr. Hillenaar:

I'm pleased to inform you that your manuscript has been deemed suitable for publication in PLOS ONE. Congratulations! Your manuscript is now with our production department. 

Kind regards, 

on behalf of

Dr Elizabeth McGill 

Academic Editor

PLOS ONE